# Controlled magnetic bistability of a helical non-Kekulé hydrocarbon on a Au(111) surface

Moheb Karbasiyoun [1,7], Marco Di Giovannantonio [2,3,7] ✉, Kalyan Biswas [4], David Écija [4,5], Olivier Blacque [1], Gonçalo Catarina [2], Nils Krane [2], Carlo A. Pignedoli [2], Pascal Ruffieux [2], José I. Urgel [4,5] ✉, Roman Fasel [2,6] ✉ & Michal Juríček [1] ✉

Recent advances in the synthesis of graphene fragments that possess unpaired π-electrons and display high-spin ground states have unlocked possibilities to explore exotic physical phenomena related to magnetism. The high degree of spin-delocalisation makes these non-metal-based systems ideal building blocks for the construction of chains and lattices with strongly correlated magnetic ground states, which is the main requisite for measurement-based quantum computation. In this work, we demonstrate the magnetic bistability of a diradical nanographene that allows direct spin manipulation at the single-molecule level. To this end, we make use of solution-phase synthesis and tip-induced activation on a metallic surface to construct a helical non-Kekulé hydrocarbon spin switch, with a reversible transformation between a magnetic ground state and a non-magnetic one via intramolecular bond formation/breaking. The switching process is monitored by scanning tunnelling spectroscopy measurements, illustrating that this, and related systems, hold potential as spin-switch units for direct manipulation of magnetism and quantum information in entangled spin systems.

Magnetic ground states in molecular graphene fragments, referred to as nanographenes, emerge from specific conjugation topologies[1,2], which give rise to an entire family of structures termed non-Kekulé[3,4], with unique electronic and magnetic properties compared to traditional Kekulé structures. The direct correlation between structure and ground-state spin enables precise control over π-electron magnetism through molecular design[5,6]. The emergence of π-magnetism has been reported to arise from sublattice imbalance[7,8], topological frustrations[9] and the interplay between hybridisation energy and Coulomb repulsion[10]. In contrast to systems with mostly localised unpaired electrons, such as in metal complexes[11], the unpaired electrons in non-Kekulé nanographenes are highly delocalised[12] within the molecular unit. This feature makes this class of carbon-based compounds ideal

building blocks to construct covalently linked oligomeric arrays with strongly correlated magnetic ground states[13] useful for quantum computation[14]. The prototypical class of such systems are triangular polybenzenoid hydrocarbons[15], exemplified by the first two homologues, phenalenyl **2 T**[16] and triangulene **3 T**[17], as shown in Fig. 1a. These nanographene π-radicals adhere to Hund's rule and possess ground states of the highest possible spin multiplicity ($S = 1/2$ and 1 for **2 T** and **3 T**, respectively), as predicted by Ovchinnikov[18] and Lieb[1]. Despite their high reactivity[19,20], several triangular nanographenes have been successfully synthesised and characterised: five pristine homologues, recently achieved[7,8,21–23] by means of on-surface synthesis and tip-induced atomic manipulation, and kinetically stabilised derivatives of the first two homologues, accomplished[24–26] via solution-phase

[1]Department of Chemistry, University of Zurich, Zurich, Switzerland. [2]nanotech@surfaces Laboratory, Empa, Swiss Federal Laboratories for Materials Science and Technology, Dübendorf, Switzerland. [3]CNR - Istituto di Struttura della Materia (CNR-ISM), Roma, Italy. [4]IMDEA Nanoscience, Madrid, Spain. [5]Unidad de Nanomateriales avanzados, IMDEA Nanoscience, Unidad asociada al CSIC por el ICMM, Madrid, Spain. [6]Department of Chemistry, Biochemistry and Pharmaceutical Sciences, University of Bern, Bern, Switzerland. [7]These authors contributed equally: Moheb Karbasiyoun, Marco Di Giovannantonio.
✉e-mail: marco.digiovannantonio@cnr.it; jose-ignacio.urgel@imdea.org; roman.fasel@empa.ch; michal.juricek@chem.uzh.ch

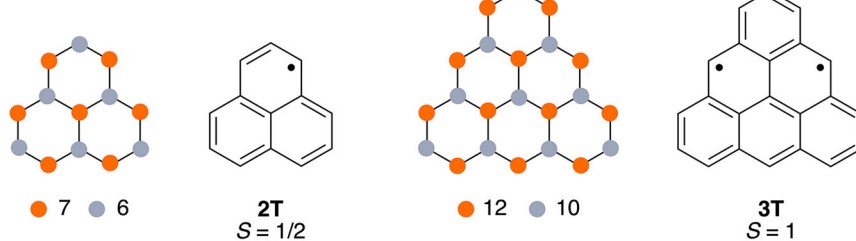

**a  nanographene π-magnetism**
from sublattice imbalance

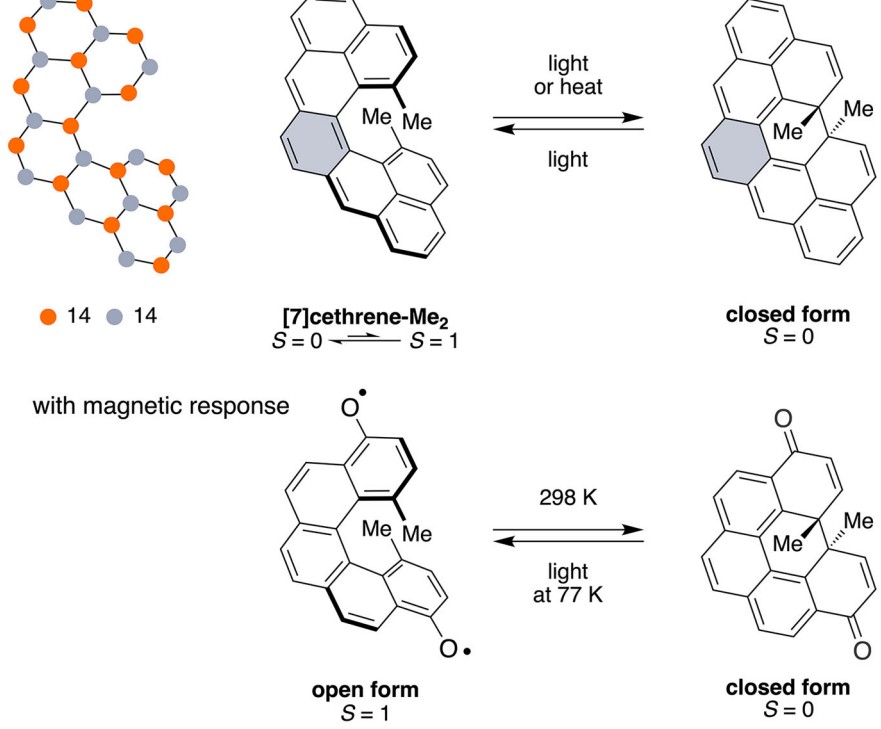

**b  solution-phase photoswitches**
without magnetic response

with magnetic response

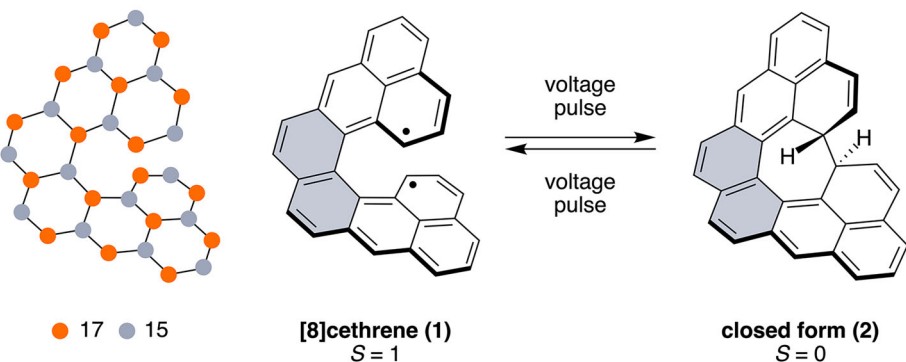

**c  single-molecule spin switch**
this work, on surface

**Fig. 1 | Molecular design of a non-Kekulé spin switch. a** Triangular non-Kekulé nanographenes **2 T** and **3 T**. Helical nanographene switches cethrene (**b**, top; dimethyl derivative shown) and [8]cethrene (**c**) featuring two **2 T** units fused in a Kekulé and non-Kekulé mode, respectively, along with Dumele's [5]helicene switch (**b**, bottom). Orange and grey dots illustrate sublattice (im)balance according to Ovchinnikov's rule, and grey filling highlights benzene and naphthalene bridging moieties.

synthesis. Both approaches provided direct evidence for the magnetic ground state of the first two species, **2 T** and **3 T**.

Using on-surface scanning probe techniques such as non-contact atomic force microscopy (nc-AFM), and scanning tunnelling microscopy (STM) and spectroscopy (STS), **2 T** and **3 T** were studied as building blocks or subunits of more complex structures, including dimers[10,27], chains[13,28], sheets[9,29], and macrocycles[30,31]. These systems—so far limited exclusively to planar structures—enabled the realisation of non-trivial magnetic ground states[32] as well as observation of spin excitations and edge fractionalisation[13]. To allow controlled manipulation of quantum information, it is essential to devise a structural unit that displays directed magnetic bistability[33]; that is, its spin can be changed on demand. Recently, a non-directed switching between π-diradical open-shell and closed-shell states in indeno[1,2-*a*]fluorene molecules on an insulating surface was reported[34]; however, it was not possible to control the formation of one or the other species, when changing the adsorption site on NaCl.

Under this scenario, a suitable molecular platform to realise such a function was lately discovered, namely, cethrene[35] (Fig. 1b, top), which belongs to a class of polybenzenoid helical hydrocarbons formally built from two **2 T** units (white) fused to an aromatic unit (grey). Their most characteristic feature is the alteration of the electronic structure between quinoidal Kekulé and diradical non-Kekulé forms, with each additional benzenoid ring expanding the helical backbone[36]. The first homologue in this series, Kekulé [7]cethrene, possesses a singlet ground state ($S = 0$) and a low-energy triplet excited state whose thermal population is dictated by the magnitude of the energy gap between the two spin states[37]. Using a dimethyl derivative of this homologue, [7]cethrene-Me$_2$, it has been demonstrated[38] that reversible ring closure of the cethrene scaffold can be promoted by light or heat in solution (Fig. 1b, top). Although this switch did not produce a magnetic response, as the triplet state of [7]cethrene-Me$_2$ is thermally inaccessible at room temperature, an analogous magnetic photoswitch operating at cryogenic temperatures was later reported[39] by Dumele and coworkers, validating the concept (Fig. 1b, bottom).

Here, we present the second homologue in the cethrene series, a helical non-Kekulé diradical with a triplet ground state—[8]cethrene (Fig. 1c). Using a STM tip-induced bond formation/breakage, this molecular system can be switched controllably and reversibly between two distinct structures: an open-shell form (**1**) with ferromagnetically coupled spins ($S = 1$) and a closed-shell form (**2**), therefore behaving as a spin switch, as we experimentally demonstrate by means of low-temperature STM and STS.

On metal surfaces, the magnetic ground state of open-shell species typically manifests as zero- or low-bias features in the STS spectra[40] that can be assigned to the total spin of the system according to the observed line shape. The appearance/disappearance of such spectral features (e.g., zero-bias Kondo resonance[41]) can thus be attributed to a switching behaviour between magnetic/non-magnetic states. Up until now, the on–off switching of the Kondo resonance has been achieved by covalent bonding to the surface[42] or to a hydrogen atom[43], intra-[44] and intermolecular[45] conformational changes, geometrical distortion[46], magnetic field[47], electron-injection-induced coordination[48], and variation of a tip–sample distance[49]. In addition, a molecular switch based on Bergman cyclisation of a diyne system on NaCl/Cu(111) was described, although the on–off switching was tracked via the observation of structural changes, and no low-bias spectroscopy was reported[50]. In our system, the appearance/disappearance of an under-screened Kondo resonance relies on a controlled chemical transformation that alters the chemical bonding within the molecular skeleton, making [8]cethrene a prototypic single-molecule system in which fully directed and reversible switching of the magnetic ground state is achieved via an intramolecular reaction. The switching process can be observed by monitoring such an under-screened Kondo resonance, attributed to ferromagnetic correlations

(triplet ground state, $S = 1$), that is either on or off depending on the form that the switch adopts upon the application of bias voltage or specific scanning parameter conditions with the STM tip: open = Kondo on, closed = Kondo off. Because of its spin-delocalised open-shell electronic structure, [8]cethrene is suited for implementation into nanographene-based spin chains and lattices as a spin-switch unit for controlling magnetism and quantum information.

## Results and Discussion

The target [8]cethrene was prepared by combining[51] the solution-phase synthesis (Fig. 2a) with a tip-induced activation on a metal surface (Fig. 2b). The key intermediate in our synthetic strategy was the dihydro-precursor of [8]cethrene (2*H*-**1**) comprising two hydrophenalene units fused to a naphthalene unit, synthesised starting from dialdehyde **3**[52]. In the first step, the Wittig reaction of **3** and **4** was performed to quantitatively afford intermediate **5** equipped with side chains required for the closure of the last two rings of the skeleton. Photocyclodehydrogenation of **5** was employed to construct the [6]helicene backbone, affording intermediate **6** in 56% yield. Subsequently, hydrolysis, acyl chloride formation, and Friedel–Crafts acylation provided diketo compound **7** in 66% yield over the three steps. Finally, **7** was subjected to reduction and dehydration to afford dihydro-precursor 2*H*-**1** in 32% yield, which was used for on-surface deposition. All intermediates and the target compound were characterised by nuclear magnetic resonance (NMR) spectroscopy and high-resolution mass spectrometry (HRMS); the structures of **7** and 2*H*-**1** are additionally supported by single-crystal X-ray diffraction analysis (XRD; see Supplementary Information), both revealing the fully assembled helical skeleton composed of eight six-membered rings.

To probe whether [8]cethrene can be generated in solution, 2*H*-**1** was oxidised with *p*-chloranil in toluene under oxygen-free conditions, leading to the formation of a precipitate within 30 minutes. On exposure of this suspension to air, diketo compound **8** was isolated, the structure of which was confirmed by NMR spectroscopy and XRD analysis (see Supplementary Information). These observations suggest that oxidation of 2*H*-**1** generates reactive mono- and/or diradical species, which form under the reaction conditions insoluble σ-oligomers. Upon exposure to air, these oligomers may transform into **8** either directly or via dissociation into monomeric mono- and/or diradical species. However, because the formation of free [8]cethrene in solution could not be directly confirmed and the involvement of mono-radical intermediates cannot be excluded, its solution-phase generation remains inconclusive.

To investigate the electronic properties and switching behaviour of [8]cethrene, the dihydro-precursor 2*H*-**1** was deposited onto a clean Au(111) surface at room temperature under ultra-high vacuum (UHV) conditions. At low molecular coverage, after cooling the sample down to 4.7 K, the molecules are found as isolated species, preferably adsorbed at the elbow site of the 22×√3 ("herringbone") surface reconstruction. While still offering interesting details to unravel the general picture[9], molecular adsorption on such under-coordinated sites[53–56] may affect the electronic properties of the species under investigation. Therefore, lateral manipulation of the molecules away from these reactive sites was routinely performed with the STM tip to reduce the interaction with the substrate and access the intrinsic electronic properties of the studied species (Supplementary Fig. 18). An overview of the observed structures on the Au(111) surface upon deposition of 2*H*-**1** and after applying different bias voltages from the STM tip is shown in Fig. 2b. It should be noted that the observed transformations were not always occurring in a stepwise manner according to the sequence shown in the chemical sketch of Fig. 2b, but an event promoted by applying voltage could entail more than one reaction step, as shown by the corresponding arrows (e.g., direct transformation of 2*H*-**1** to **1**). The DFT-computed STM images of all the structures adsorbed on the Au(111) surface offered excellent

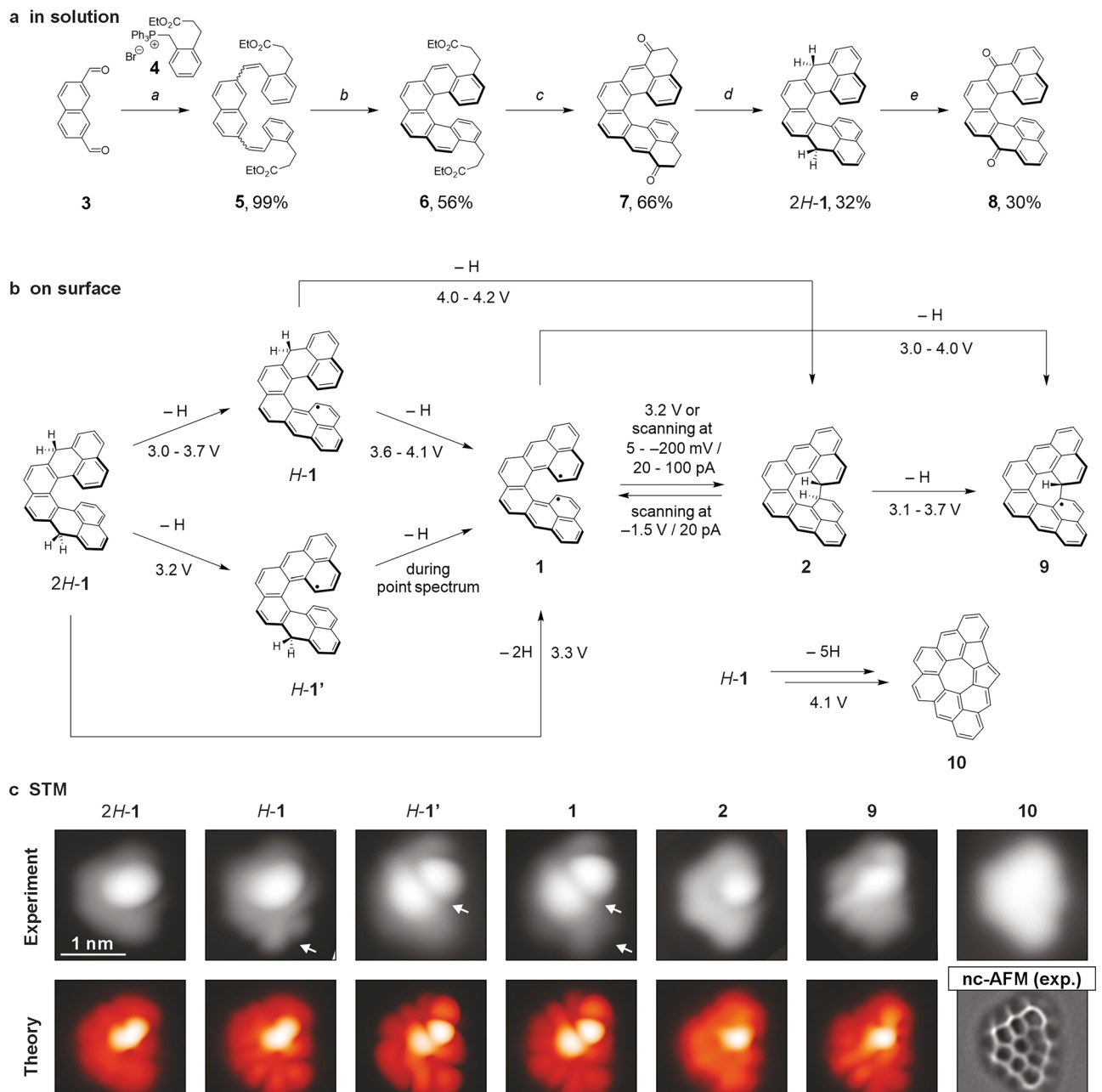

**Fig. 2 | Synthesis of dihydro-precursor 2H-1 and imaging of the species generated from it on Au(111). a** Reaction conditions: a) EtONa, THF, −20 °C to rt, b) hv, I₂, 2-methyloxirane, toluene, c) i) LiOH, THF/water (4:1), rt, ii) (COCl)₂, 65 °C, iii) AlCl₃, CH₂Cl₂, −78 to −30 °C, d) i) NaBH₄, CH₂Cl₂/EtOH (2:1), rt, ii) p-TSA, toluene, 90 °C, e) i) p-chloranil, toluene, rt, ii) air, toluene, rt. p-TSA = p-toluenesulfonic acid. **b** Chemical sketch of the structures observed on the Au(111) surface upon deposition of 2H-1 and applying voltage from the STM tip (see Method section for experimental details and voltage application procedure). Arrows indicate the direction of the observed transformations and the numbers next to the arrows specify the voltage pulses or scanning conditions at which the transformation was observed (ranges arise from multiple experiments). **c** Experimental and DFT-calculated STM images of all observed species (see Supplementary Fig. 23 for the optimised geometries). Scanning parameters: $V_b$ = −0.2 V, $I_t$ = 20 pA for all STM images. In the case of **10**, nc-AFM imaging was performed (bottom-right). Scanning parameters: Δz = +185 Å with respect to STM set point: $V_b$ = −5 mV, $I_t$ = 100 pA. All STM and nc-AFM images have the same size.

agreement with the experimental images, thus confirming our assignment of the different species (Fig. 2c, experimental and theoretical STM images).

The pristine precursor 2H-1 displays a characteristic trapezoidal shape with a bright lobe non-symmetrically displaced with respect to the molecular centre, representing the upper end of the helical moiety that is more distant from the surface plane (Fig. 2c). Annealing of the as-prepared sample until complete molecular desorption did not induce any intramolecular chemical transformations. Thus, a voltage pulse (see Methods) from the STM tip was applied to induce dehydrogenation[57] of 2H-1 in an attempt to achieve the target 8cethrene. Initially, monoradical species H-1 and H-1' were obtained, with one hydrogen atom being removed in each case from the lower or upper half of the molecule, respectively. Experimentally, such events result in the appearance of characteristic lobes at one of the two ends of the molecule (white arrows in the STM images of H-1 and H-1' in Fig. 2c). Further hydrogen atom removal from H-1 and H-1' was achieved by applying a stronger voltage pulse to afford species **1**, which could also be accessed directly from the precursor 2H-1. The STM image of **1** features two pairs of lobes, one at the lower and one at

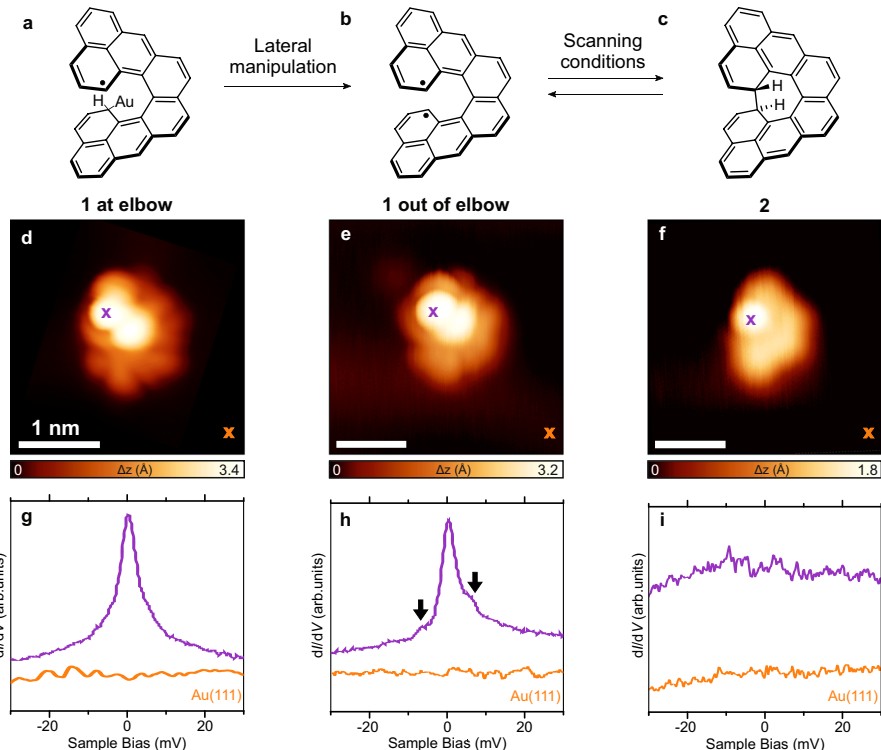

**Fig. 3 | Electronic characterisation of the [8]cethrene spin switch. a–c** Chemical structures of the three investigated species. When **1** is at the elbow site of the herringbone reconstruction of Au(111), the lower radical site is quenched due to a bonding interaction with the gold substrate, depicted as a bond with Au in (**a**). **d–f** High-resolution STM images acquired with a CO-functionalized tip of species **1** located at the elbow site, out of it, and species **2**. Scanning parameters: $V_b$ = 0.05 V, $I_t$ = 50 pA (**d**), $V_b$ = −1.5 V, $I_t$ = 20 pA (**e**), $V_b$ = −0.2 V, $I_t$ = 20 pA (**f**), scale bars = 1 nm. **g–i** d$I$/d$V$ spectra acquired for the three species at the locations indicated by the crosses in the STM images and showing marked differences in the line shapes, as discussed in the text. Open-feedback parameters for the d$I$/d$V$ spectra: $V_b$ = 50 mV, $I_t$ = 0.5 nA; root-mean-squared modulation voltage $V_{rms}$ = 0.8 mV. Reference spectra shown in (**g–i**) were taken on the bare Au(111) surface (orange lines).

the upper end of the molecule, indicating the formation of the target diradical [8]cethrene (**1**). In some cases, when attempting the dehydrogenation of specie $H$-**1** to **1**, a skeletal rearrangement accompanied by the loss of five hydrogen atoms was observed, affording planar specie **10**. The planarity of the molecule facilitated us to confirm its chemical structure by nc-AFM measurements using a CO-functionalized tip, which was not possible for all previous, highly non-planar species. The resulting constant-height frequency-shift image of **10** features two five- and one seven- in addition to six-membered rings.

To obtain information about the electronic nature of **1**, we performed low-bias d$I$/d$V$ spectroscopy (Fig. 3). We observed two types of spectroscopic features, one for molecules located on the elbows of the Au(111) herringbone reconstruction (Fig. 3a, d, g) and another one for molecules located in other areas of the Au(111) surface (Fig. 3b, e, h). When **1** is located on the elbow, we observe a spectroscopic feature centred at the Fermi level, which we attribute to a narrow Kondo resonance (FWHM ~3 meV), consistent with an unpaired electron spin ($S$ = ½) interacting with the electron bath of Au(111) (Fig. 3g). This finding suggests that **1** might be pinned to the elbow site of the herringbone ridge, possibly binding to an under-coordinated gold atom[9,53–56], which could lead to a reduction in the number of unpaired electrons from two to one (see Fig. 3a for the corresponding chemical sketch).

The spectrum on **1** located in other areas (out of elbow) of the Au(111) reconstruction reveals a subtle but noticeable difference with respect to **1** pinned to the elbow (Fig. 3h). Herein, a Kondo feature is also observed. However, this feature is now accompanied by two small and symmetric shoulders around the Fermi energy (Fig. 3h). These shoulders can be attributed to spin-flip features suggesting

ferromagnetic correlations (triplet ground state, $S$ = 1) with an exchange coupling energy of $J_{eff}$ ~8 meV. The emergence of magnetism in **1** is qualitatively reproduced by gas-phase calculations via mean-field Hubbard (MFH, Supplementary Fig. 24) and DFT, with a triplet ground state preferred over a singlet antiferromagnetic one by a few meV (7 meV by MFH and 20 meV by DFT). Therefore, this underscreened $S$ = 1 Kondo effect implies that, when **1** is away from the elbow site, it is no longer pinned to the gold surface with one unpaired electron being "quenched", and the two unpaired electrons in **1** interact ferromagnetically with one another. Interestingly, the detection of such spin-flip features in molecules with a triplet ground state and small exchange coupling energies explored on metal surfaces by STS measurements is often challenging. Rather small magnetic fields[58] or fittings using the Frota function[21,59] have already been reported to confirm their magnetic nature. However, in the present case, and other reported ones displaying relatively large exchange coupling energies[60], the spin-flip features can be directly detected by STS. Finally, to discard any mechanical manipulation or influence of the spin state caused by the STM tip while scanning over the molecule, we have acquired d$I$/d$V$ spectra measured at different tip heights. Retracting or approaching the tip ± 200 pm with respect to the starting position did not give rise to any noticeable modification of the d$I$/d$V$ spectra (Supplementary Fig. 19), while approaching closer than 200 pm leads to the lateral displacement/desorption of **1**.

Next, we addressed the challenge of reversible intramolecular transformation with the aim of switching the magnetic character of **1** off and then turning it on again. To our delight, the switching was achieved by applying a voltage of 3.2 V to **1** or by scanning over the molecule with very low bias voltages ($V_b$ = 5 to −200 mV, $I_t$ = 20 to 100 pA). These

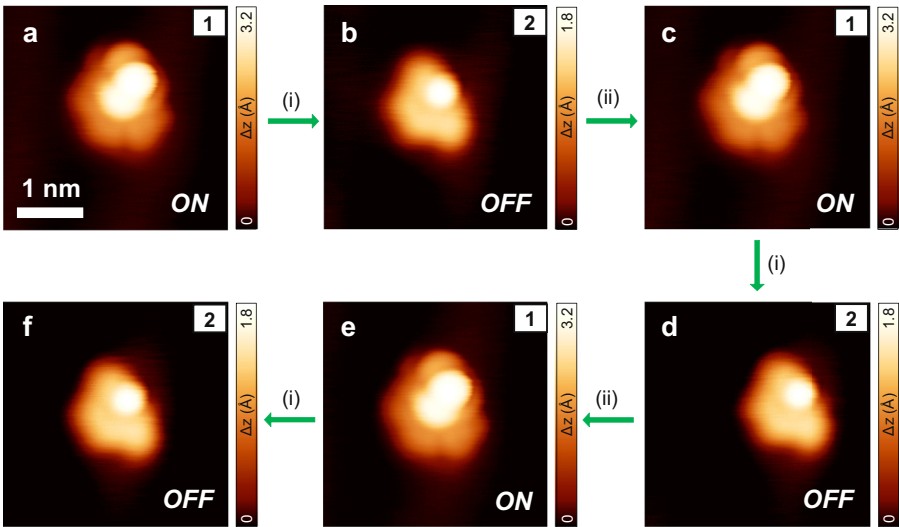

**Fig. 4 | Reversibility of the [8]cethrene spin switch. a–f** Selected sequence of molecular transformations from **1** to **2** and vice versa, enabled by specific scanning conditions: (i) $V_b$ = 5 mV, $I_t$ = 20 pA, (ii) $V_b$ = −1.5 V, $I_t$ = 20 pA. Scanning parameters: $V_b$ = −1.5 V, $I_t$ = 20 pA (**a, c, e**), $V_b$ = −0.2 V, $I_t$ = 20 pA (**b, d, f**). Scale bars = 1 nm.

specific experimental conditions promoted the transformation of **1** into **2** (Fig. 4). During this intramolecular process, a new C–C bond is formed, which alters the electronic configuration of the molecule and renders it formally a closed-shell structure (Fig. 3c, f, i). Indeed, low-bias d$I$/d$V$ spectroscopy of **2** did not reveal any features that could be ascribed to the presence of unpaired electrons (Fig. 3i). Notably, this process was found to be reversible, and species **2** could be transformed back into species **1** by scanning over the molecule with a large bias ($V_b$ = −1.5 V, $I_t$ = 20 pA). The energy barrier to convert **1** into **2** (the latter being more stable by 0.08 eV) is 0.29 eV, while that of the reverse transformation is 0.37 eV, as computed by climbing image nudged elastic band (CI-NEB) calculations[61]. The switching from species **1** to **2** and vice versa was experimentally tested 40 times (20 times transforming **1** into **2** and 20 times transforming **2** into **1**). The switching was successfully achieved in all cases when induced during the scan, with a 100% yield. When applying a higher voltage instead, the result of the transformation was sometimes different; namely, a hydrogen atom was removed from **2**, yielding species **9**, a monoradical with one hydrogen atom protruding out of the molecule (Fig. 2b). Note that scanning over **1** or **2** using intermediate voltage parameters (from 0.1 V to 1.4 V and from −0.2 V to −1.4 V, with $I$ = 20 pA) does not give rise to any molecular transformation. An illustrative sequence of the reversible switching process of [8]cethrene, including the employed scanning parameters, is shown in Fig. 4 (complete series of seven transformations is shown in Supplementary Fig. 20).

We also notice that the molecule under investigation often moved during the transformation from **1** to **2** (i.e., it slightly changed its adsorption location on the Au(111) surface, see Supplementary Fig. 21) in both cases of "voltage pulses" and the "scanning method". In contrast, the molecule preserved its adsorption location during the transformation of **2** into **1**. Finally, in one case, we observed the transformation from **1** to **2** during the lateral manipulation of specie **1** (while dragging the molecule out of an elbow site). All these observations suggest the following scenario for the reversible switching between **1** and **2**: In both cases of "voltage pulse" or "scanning method", molecule **1** receives a mechanical excitation that makes it relax into specie **2**, which is slightly more stable than **1** (by 0.08 eV; as computed by DFT for the two species adsorbed on Au(111)). On the other hand, specie **2** can be converted back into **1** while scanning at bias voltages close to the energy of its frontier orbitals (i.e., −1.5 V, as shown in Supplementary Fig. 22), suggesting a mechanism based on hole-injection into the HOMO.

We have demonstrated the directed magnetic bistability in a non-Kekulé nanographene diradical, where reversible switching of the ground-state spin−from triplet ($S$ = 1) to singlet ($S$ = 0)−is enabled through intramolecular ring-closing reaction promoted by tip-induced voltage pulses at the helical [8]cethrene molecule. This hydrocarbon spin switch was achieved by combining a solution-phase synthesis employed to construct the dihydro-precursor with a stepwise tip-induced activation on a metallic surface to yield mono- and diradical species in two consecutive dehydrogenation steps. The low-bias d$I$/d$V$ spectra of the open diradical form of the switch located outside the Au(111) elbow sites show a Kondo feature that is accompanied by a spin-flip signature attributed to an under-screened spin triplet, in agreement with theoretical calculations. Upon applying a bias voltage pulse of 3.2 V or specific scanning conditions, the open-shell form undergoes an intramolecular transformation to the closed-shell form, during which a new bond and a new ring are formed. The low-bias d$I$/d$V$ spectra of the closed-shell form do not display any features that would indicate the presence of unpaired electrons, which is in accord with the closed-shell electronic structure of this species. This switching process was found to be fully reversible. In light of the recent developments enabling bottom-up construction of nanographene-based multispin arrays, this work puts forward a tool for direct manipulation of spin and thereby brings prospects for applications in the areas of carbon magnetism and quantum information processing.

## Methods

### Solution-phase synthesis
Synthetic protocols and characterisation data for all compounds are compiled in the Supplementary Information.

### X-ray crystallography
Single crystals for all three compounds were obtained from the corresponding $CD_2Cl_2$ solutions by slow evaporation of the solvent in an NMR tube under a stream of nitrogen. Single-crystal X-ray diffraction data were collected at 160(1) K on a Rigaku OD SuperNova/Atlas area-detector diffractometer using a single-wavelength X-ray source (Cu Kα radiation: $\lambda$ = 1.54184 Å[62]) or an Oxford Instruments Cryojet XL cooler. The selected suitable single crystal was mounted using polybutene oil on a flexible loop fixed on a goniometer head and immediately transferred to the diffractometer. Pre-experiment, data collection, data reduction and analytical absorption correction[63] were performed with the programme suite CrysAlisPro[64]. Using Olex2[65], the structure was

solved with the SHELXT[66] small-molecule structure-solution programme and refined with the SHELXL 2018/3 programme package[62] by full-matrix least-squares minimisation on $F^2$. PLATON[67] was used to check the result of the X-ray analysis. For more details about the data collection and refinement parameters, see the CIF files. Crystal parameters are available in the Supplementary Information.

## On-surface experiments

Synthesis and measurements were performed under UHV conditions in two independent custom-designed ultra-high vacuum systems (base pressure below $2 \times 10^{-10}$ mbar) hosting commercial low-temperature microscopes with STM/AFM capabilities from Scienta Omicron. Au(111) substrates (MaTeck GmbH) were cleaned by repeated cycles of Ar$^+$ sputtering (1 keV) and annealing (470 °C). The precursor molecules were thermally evaporated onto the clean Au(111) surface from quartz crucibles heated at 165 °C, which resulted in a deposition rate of ~0.5 Å·min$^{-1}$. STM images were acquired at 4.7 K in constant-current mode using an etched tungsten tip. Bias voltages are given with respect to the sample. nc-AFM measurements were performed at 4.7 K with a tungsten tip placed on a qPlus tuning fork sensor[68]. The tip was functionalized with a single CO molecule at the tip apex picked up from the previously CO-dosed surface[69]. The sensor was driven at its resonance frequency (24800 Hz) with a constant amplitude of 70 pm. The frequency shift from resonance of the tuning fork was recorded in constant-height mode using Omicron Matrix electronics and HF2Li PLL by Zurich Instruments. The $\Delta z$ is positive (negative) when the tip-surface distance is increased (decreased) with respect to the STM set point at which the feedback loop is opened.

In our work, we have used two types of external stimuli to achieve the molecular transformations: (i) "voltage pulses" (i.e., bias ramping with the tip at a constant height, far from the molecule, as described below) and (ii) "scanning method", i.e., specific scanning conditions during constant-current STM image acquisition. While voltage pulses have been successfully used for all molecular transformations depicted in Fig. 2, they could not induce the transformation of **2** into **1** (i.e., the reversible switching described in our manuscript). In this case, the only successful strategy found by us was the "scanning method" (−1.5 V/20 pA). Regarding the voltage applications reported in Fig. 2, they are not standard pulses applied within a short time frame, but performed by placing the tip on top of the molecule and ramping up the bias voltage until an event occurs. This method guarantees more controlled and reproducible modifications and assessing of voltage thresholds. Standard procedure is as follows: (i) stop the tip on top of the molecule – in the centre, as the electric field will interest the entire molecule – with scanning parameters $V_b = 0.1$ V, $I_t = 10$ pA; (ii) open the feedback loop; (iii) increase $\Delta z$ of +0.5 nm (tip moves away from the molecule) such that the tunnelling current is almost zero; (iv) slowly ramp up the bias voltage while monitoring the current-vs-time signal: a sudden jump indicates an event, which typically occurs within a few seconds after reaching of the required voltage; (v) ramp down the bias to $V_b = 0.1$ V, $I_t = 10$ pA; (vi) close the feedback-loop; (vii) restart the scan.

The d$I$/d$V$ measurements were obtained with a lock-in amplifier operating at a frequency of 800 Hz. Modulation voltages for each measurement are reported as root-mean-squared amplitude ($V_{rms}$). Tunnelling bias voltages are given with respect to the sample.

## Theory

All DFT calculations were performed using the AiiDAlab platform[70] and AiiDA workflows[71] based on the CP2K code[72]. The surface–adsorbate systems were modelled in the repeated slab scheme. The simulation cell consisted of four atomic layers of gold along the [111] direction. A layer of hydrogen atoms was used to passivate one side of the slab to suppress the Au(111) surface state. A vacuum of 40 Å was included in the simulation cell to decouple the system from its periodic replicas in

the direction perpendicular to the surface. The electronic states were expanded using a TZV2P Gaussian basis set[73] for carbon and hydrogen species and a DZVP basis set for gold species. A cutoff of 600 Ry was used for the plane-wave basis set. Norm-conserving Goedecker–Teter–Hutter pseudo-potentials[74] were used to represent the frozen core electrons of the atoms. We used the Perdew–Burke–Ernzerhof parameterisation for the generalised gradient approximation of the exchange-correlation functional[75]. To account for van der Waals interactions, we used the D3 scheme proposed by Grimme[76]. The gold surface was modelled using a super-cell, with a size of $41.3 \times 40.8$ Å$^2$ (896 gold atoms corresponding to 224 surface unit cells). The Au(111) slab was planar, and the herringbone reconstruction associated with this surface was not considered, as it would greatly expand the supercell and not substantially change the chemical activity of the surface[77]. To obtain the equilibrium geometries, we kept the atomic positions of the bottom two layers of the slab fixed to the ideal bulk positions, and all other atoms were relaxed until forces were lower than 0.005 eV/Å. STM images were simulated within the Tersoff–Hamann approximation[78] based on the Kohn–Sham orbitals of the slab/adsorbate systems. The orbitals were extrapolated to the vacuum region in order to correct the wrong decay of the charge density due to the localised basis set. To compute the triplet/singlet energy difference in the gas phase, we used the geometry of the adsorbed molecule and unrestricted Kohn–Sham DFT, starting from a ferromagnetic (antiferromagnetic) guess and imposing a multiplicity of 3 (1). From the same geometries, we built a tight-binding model considering one orbital and one electron per carbon atom. Given the non-planarity of the molecules considered, we computed the hoppings $t_{ij}$ between carbon sites $i$ and $j$ using a Slater–Koster formalism for the angular dependence, and assuming an exponential decay with the distance (see ref. 79 for details and parameters). In the mean-field Hubbard routine, we considered an on-site Hubbard repulsion of $U = 3$ eV, and a tolerance of $10^{-6}$ for the convergence of the spin-resolved densities in the self-consistent loops.

## Data availability

All data associated with this study are available in the published Article and its Supplementary Information. The raw NMR and IR data generated in this study have been deposited in the public repository Zenodo under the accession link https://zenodo.org/record/8253265 (https://doi.org/10.5281/zenodo.8253265). The X-ray crystallographic coordinates for structures reported in this study have been deposited at the Cambridge Crystallographic Data Centre (CCDC), under deposition numbers 2289143 (**7**), 2289144 (2*H*-**1**) and 2446997 (**8**). These data can be obtained free of charge from The Cambridge Crystallographic Data Centre via www.ccdc.cam.ac.uk/data_request/cif. The on-surface data (experimental and computational, including Cartesian coordinates) generated in this study have been deposited in the Materials Cloud repository under the accession link https://doi.org/10.24435/materialscloud:ae-s0. All data are available from the corresponding authors upon request.

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

## Acknowledgements

This project received funding from the European Research Council (ERC) under the European Union's Horizon 2020 research and innovation programme (grant agreement number 716139, M.J.), the Swiss National Science Foundation (SNSF; PP00P2_170534 (M.J.), PP00P2_198900 (M.J.), 200020_212875 (R.F.) and CRSII5_205987 (P.R. and M.J.)) and the NCCR MARVEL, a National Centre of Competence in Research, funded by the Swiss National Science Foundation (grant number 205602, C.A.P.). C.A.P. thanks the Swiss Supercomputing Center (CSCS) for computational support (project s1267, lp83). G.C. and R.F. acknowledge financial support from the Werner Siemens Foundation (CarboQuant). We are grateful to Prof. Tomáš Šolomek (University of Amsterdam) for helpful discussions and Prof. Peter Štacko (University of Zurich) for creating a graphical abstract, which, unfortunately, could not be included here. We gratefully acknowledge Lukas Rotach for technical support during the experiments and Paula Widmer for acquiring the melting points and IR spectroscopy data.

## Author contributions

M.D.G., J.I.U., R.F. and M.J. conceived and supervised the project. M.K. synthesised and characterised the precursor molecules. O.B. performed the XRD measurements. M.D.G. and K.B. performed the on-surface experiments and SPM measurements with inputs from P.R. and D.E. C.A.P., G.C. and N.K. performed the DFT and MFH calculations. M.K., M.D.G. and M.J. co-wrote the manuscript, with the contribution from all authors.

## Competing interests

The authors declare no competing interests.
