## [Transparent Peer Review file · Nature Communications]

Controlled magnetic bistability of a helical non-Kekulé hydrocarbon on a Au(111) surface

Corresponding Author: Professor Michal Juríček

Version 0:

Reviewer comments:

Reviewer #1

(Remarks to the Author)

This work used solution-phase synthesis to synthesize the key precursor 2H-1, and then obtained a helical two-radical molecule-[8]cethrene with a ferromagnetically coupled ground state by tip-induced dehydrogenation on the Au(111) surface. By detecting zero-bias or low-bias characteristics in STS spectra, specifically, Kondo effect and spin-flip feature, the authors demonstrated that reversible magnetic ground state switching can be achieved by tip-induced intramolecular transformation on [8]cethrene. The manuscript is well-written and the target helical molecule-[8]cethrene is novel. However, the understanding of the mechanism of tip manipulation deserves further exploration. More experimental data and mechanistic discussion should be supplied before it can be published in Nature Communication.

1. The experimental conditions for realizing the conversion from 1 to 2 are confusing. For voltage ramping, the bias threshold is 3.2 V, and molecule 1 also can be activated with very low bias voltages ($V = -5$ mV). Why is the bias difference between the two ways is so great? Generally, the bias voltages of electronic excitation are high, because the holes/electrons are injected into the molecular orbital. But the vibration excitation mechanism is often accompanied by a lower bias threshold (for example, the energy of C–H vibrational modes is about ~ 400 meV). Additionally, can the conversion from 2 to 1 be achieved by biasing ramping?

2. The mechanism of the tip manipulation is not clear. Referring to the reported studies, detailed statistical data of reversible experiments and manipulation mechanism should be provided. Specifically, Is tip-induced manipulation due to the electronic excitation, vibrational excitation or electric field? Is the process one electron process or a multiple-electron process? What about switching yields?

3. What are the energy barriers for bond formation/breakage between molecules 1 and 2? Whether they are related to the bias thresholds for tip manipulation?

4. Is it possible to obtain STS spectra with wide ranges of molecules 1 and 2 and dI/dV maps of corresponding peak positions, and explain them through DFT calculations?

5. Although the authors write in the Method section that the tip is placed at the center of the molecule during the tip manipulation, no indication is marked in Figure 4. The site of manipulation on molecules 1 and 2 should be clearly indicated. In addition, have the authors attempted manipulation at different positions of the molecule 1 and 2 and does their voltage thresholds change? Regardless of whether it changes or not, manipulation experiments on different positions would benefit the understanding of the manipulation mechanism.

6. In Page 8, line 194, the authors mentioned that the exchange coupling energy of molecule 1 was calculated using DFT in gas phase. However, the detailed description of the calculation is not found in the Method section and Supporting Information.

Additionally, some minor issues should be noticed:

1. Page 6, line 137. The subgraphs in Figure 2 were not labeled with a, b, c....
2. Page 8, line 176. The values of scale bars should be provided, although they are same.

Reviewer #2

(Remarks to the Author)

In this manuscript, the authors report the new non-Kekulé hydrocarbon [8]cethrene and its on surface synthesis. The hydrogenated precursor is prepared in solution, deposited on an Au(111) surface and dehydrogenated applying a voltage. The subsequent chemical reaction sequence is studied in detail using scanning tunnelling microscopy (STM) and further supported by theoretical computations. The dehydrogenated target diradical 1 can be reversibly converted to a closed form non-radical molecule 2 by applying different scanning voltages. Further dehydrogenations from 2 were also observed and thoroughly characterized.

Overall, the manuscript is at a high level of scientific presentation. For an expert reader, it is well-written and clearly understandable. The characterization of all compounds synthesized in solution is conclusive and partially unambiguous structural proof is provided by single-crystal X-ray diffraction. The on-surface characterization is conclusive but could perhaps be extended using more specialized techniques to express the magnetic key features of the molecule, such as EPR-sensitive STM (Physical Review Research 2, 013032 (2020)). However, this could be done in a follow-up study and is not required to publish the current manuscript.

In summary, I highly recommend publication after minor revisions of some points as listed below. Congratulations to the authors on this fantastic piece. This study closes the gap of on-surface spin state switching with all-organic molecules (see my comments below).

1: It remains open whether the authors have attempted the full solution-phase synthesis of the targeted structures 1 or 2 (or 8), starting from 2H-1 or 7? What experiments were conducted (conditions, outcome, observations), and did any of the attempts lead to similar side products, such as compound 9?

2: The citation of references 39 (and 40) is misleading when describing the [7]cethrene-Me₂ case. It should be clarified that these references refer not to dimethyl[7]cethrene but to a different spin-state switch. The difference in structure and properties should also be shown in Figure 1 to give context of the current design. I admit this is the previous work of this reviewer and typically I refrain from suggesting references to my own work during the peer review process. However, it might be the most relevant context to this study.

3: Previous work on spin state switching on an Ag-surface based on an organic ligand system on a metal-porphyrin was reported. The switching trigger was the injection of an electron. The paper should be cited: Nature Nanotechnology 2020, 15, 18–21. Their read-out was also based on the Kondo resonance.

4: The solvent system to obtain single-crystal of 2H-1 is inconsistent. In figure 2 (top right); CH₂Cl₂ and in X-ray crystallography methods CD₂Cl₂.

5: I highly suggest specifying the title and mentioning the “on surface” study in the title. The field of surface science is clearly distinguished and many phenomena that can be observed on surface are practically impossible to achieve in bulk solution. The current study is one such example and it should be reflected in the title, maybe as “On-surface controlled magnetic bistability of a helical non-Kekulé hydrocarbon” or “Controlled magnetic bistability of a helical non-Kekulé hydrocarbon on Au(111) surface”.

Reviewer #3

(Remarks to the Author)

The authors report the synthesis of an interesting molecule revealing a bistability as a diradical and as a closed-shell structure. The molecules are characterized on a Au(111) surface under vacuum by STM, whose spectroscopic variant allows probing magnetic fingerprints of the molecules. The switching is triggered by the STM current, allowing for an in-situ characterization of the reversible switching process. The results are very interesting and the data convincing. I thus consider the manuscript appropriate for publication, provided the authors address the following issues:

- Although the Introduction and Conclusion sections are labelled, there is no label for Results, which is very confusing in the first reading, knowing this is not a letter.
- As the authors describe in the main text, there have been many reports on the on and off switching of Kondo resonances by various means. In this respect, to really warrant sufficient novelty to this work, they should show that the switching is really controllable and at least provide a better characterization of it (e.g. switching traces or average life time as a function of bias, switching probability as a function of bias and/or current...). Right now it is extremely vaguely described, providing three “switching conditions” (3.2 V pulse, U=–5mV/I=20pA scan and U=–1.5V/I=20pA scan) that are neither compared nor quantitatively analyzed by any means.
- The assignment of 1 at the herringbone elbows to S=1/2 is not fully convincing. The tails around the Kondo resonance actually cause an impression as if they could be made up by the same lateral side-peaks/excitations as out-of-elbow, only a little bit broader or worse resolved. The authors should discuss this option.

Version 1:

Reviewer comments:

Reviewer #1

(Remarks to the Author)

The long-range STS of species 1 and 2 were provided in the revised SI. The results reveal a significant difference in term of HOMO-LUMO energy gap and their energy level. For instance, the HOMO position shifts nearly 1 eV lower from specie 1 to specie 2. How does this experimental finding compare with the theoretical results?

Reviewer #2

(Remarks to the Author)

The authors present a fully revised version of their manuscript. All my comments have been addressed perfectly. I want to congratulate the authors on these results. Great job.
I recommend publication without further changes.

Reviewer #3

(Remarks to the Author)

The authors have satisfactorily answered my comments and I find the article now acceptable for publication.

Version 2:

Reviewer comments:

Reviewer #1

(Remarks to the Author)

The revision is recommended for publishing

Response to Reviewer's Comments

Reviewer 1

1. The experimental conditions for realizing the conversion from **1** to **2** are confusing. For voltage ramping, the bias threshold is 3.2 V, and molecule **1** also can be activated with very low bias voltages ($V = -5$ mV). Why is the bias difference between the two ways is so great? Generally, the bias voltages of electronic excitation are high, because the holes/electrons are injected into the molecular orbital. But the vibration excitation mechanism is often accompanied by a lower bias threshold (for example, the energy of C–H vibrational modes is about ~ 400 meV).

We appreciate the reviewer's comment. We understand that the conditions required for the conversion from **1** to **2** and vice versa may need further clarification. First, we would like to stress that our main purpose was to find conditions to achieve a reversible and controllable switching between the two species, while a detailed mechanistic insight into the processes driving the conversion is out of the scope of the present study. In our work, we have used two types of external stimuli to achieve the molecular transformations: (i) "voltage pulses" (i.e., bias ramping with the tip at constant height, far from the molecule, as described in the methods) and (ii) "scanning method", i.e., specific scanning conditions during constant current STM image acquisition. While voltage pulses have been successfully used for all molecular transformation depicted in Figure 2, they could not induce the transformation of **2** into **1** (i.e., the reversible switching described in our manuscript). In this case, the only successful strategy found by us was the "scanning method" (-1.5 V / 20 pA). After discovering this pathway, we also applied the "scanning method" to attempt the transformation from **1** to **2** and found that indeed the expected product was achieved when scanning in the range $+5... -200$ mV / 20...100 pA (updated in the new Figure 2 in our revised manuscript).

We also notice that the molecule under investigation moved during the transformation from **1** to **2** (i.e., it slightly changed its adsorption location on the Au(111) surface) in both cases of "voltage pulses" and the "scanning method". In contrast, the molecule preserved its adsorption location during the transformation of **2** into **1**. Finally, in one case, we observed the transformation from **1** to **2** during the lateral manipulation of specie **1** (while dragging the molecule out of an elbow site).

All these observations suggest the following scenario for the reversible switching between **1** and **2**: In both cases of "voltage pulse" or "scanning method", molecule **1** receives a mechanical excitation that makes it relax into the slightly more stable (by 0.08 eV) specie **2** (see the energy landscape in our reply to point 3 and Figure R1 below). On the other hand, specie **2** can be converted back into **1** while scanning at bias voltages close to the energy of its frontier orbitals (i.e. -1.5 V, as shown in our reply to point 4 and Figure R2 below), suggesting a mechanism based on hole-injection into the HOMO.

Additionally, can the conversion from **2** to **1** be achieved by biasing ramping?

No, when ramping the bias on molecule **2**, we have only observed the transformation of **2** into **8**, as indicated in Figure 2 of our manuscript.

Action: We have included a Figure in the SI (Supplementary Fig. 21) showing the displacement of molecules **1** and their subsequent transformation to **2** while scanned at certain parameters. Additionally, we have improved the description of external stimuli application to achieve the desired molecular transformations (both in the Methods and in the main text) and included a discussion of the possible mechanisms driving the switching behavior, as follows:

- In the Methods: “[...]The Δz is positive (negative) when the tip-surface distance is increased (decreased) with respect to the STM set point at which the feedback loop is opened. In our work, we have used two types of external stimuli to achieve the molecular transformations: (i) “voltage pulses” (i.e., bias ramping with the tip at a constant height, far from the molecule, as described below) and (ii) “scanning method”, i.e., specific scanning conditions during constant current STM image acquisition. While voltage pulses have been successfully used for all molecular transformation depicted in Figure 2, they could not induce the transformation of **2** into **1** (i.e., the reversible switching described in our manuscript). In this case, the only successful strategy found by us was the “scanning method” ($-1.5 \text{ V} / 20 \text{ pA}$). Regarding the voltage applications reported in Figure 2, they are not standard pulses[...]
- In the main text: “[...]An illustrative sequence of the reversible switching process of [8]cethrene including the employed scanning parameters is shown in Fig. 4 (complete series of seven transformations is shown in Supplementary Fig. 20). We also notice that the molecule under investigation often moved during the transformation from **1** to **2** (i.e., it slightly changed its adsorption location on the Au(111) surface, see Supplementary Fig. 21) in both cases of “voltage pulses” and the “scanning method”. In contrast, the molecule preserved its adsorption location during the transformation of **2** into **1**. Finally, in one case, we observed the transformation from **1** to **2** during the lateral manipulation of specie **1** (while dragging the molecule out of an elbow site). All these observations suggest the following scenario for the reversible switching between **1** and **2**: In both cases of “voltage pulse” or “scanning method”, molecule **1** receives a mechanical excitation that makes it relax into specie **2**, which is slightly more stable than **1** (by 0.08 eV ; as computed by DFT for the two species adsorbed on Au(111)). On the other hand, specie **2** can be converted back into **1** while scanning at bias voltages close to the energy of its frontier orbitals (i.e., -1.5 V , as shown in Supplementary Fig. 22), suggesting a mechanism based on hole-injection into the HOMO.”

2. The mechanism of the tip manipulation is not clear. Referring to the reported studies, detailed statistical data of reversible experiments and manipulation mechanism should be provided. Specifically, Is tip-induced manipulation due to the electronic excitation, vibrational excitation or electric field? Is the process one electron process or a multiple-electron process? What about switching yields?

We are grateful to the reviewer for this comment. Our reply to point 1 offers clarifications regarding the switching mechanisms occurring during tip manipulation. When using the methods and conditions described in our manuscript, the switching yields between **1** and **2** are found to be 100%.

Action: We have included new description and discussions, as per our reply to point 1, and added the switching yield value, as follows: “[...]The switching was successfully achieved in all cases when induced during the scan, with a 100% yield. When applying a higher voltage instead[...]”

3. What are the energy barriers for bond formation/breakage between molecules 1 and 2? Whether they are related to the bias thresholds for tip manipulation?

The energy barrier for bond formation/breakage was found to be 0.29/0.37 eV, respectively. The diagram showing the energy landscape is reported below for completeness. The energy values were obtained by performing a climbing image nudged elastic band (CI-NEB) calculations [Ref. 60].

The activation barriers for the two processes are similar and could not directly justify the difference observed in the scanning parameters used for the switching from 1 to 2 and vice versa. Instead, we provide a different argument in our reply to point 1.

Figure R1. Energy landscape for the bond formation/breakage leading to the transformation of 1 to 2 and vice versa, computed by CI-NEB calculations.

Action: We have included these values in the text:

“Notably, this process was found to be reversible, and species 2 could be transformed back into species 1 by scanning over the molecule with a large bias ($V_b = -1.5$ V, $I_t = 20$ pA). The energy barrier to convert 1 into 2 (the latter being more stable by 0.08 eV) is 0.29 eV, while that of the reverse transformation is 0.37 eV, as computed by climbing image nudged elastic band (CI-NEB) calculations⁶⁰. The switching from...”

4. Is it possible to obtain STS spectra with wide ranges of molecules 1 and 2 and dI/dV maps of corresponding peak positions, and explain them through DFT calculations?

We thank the reviewer for this valuable comment. We have acquired STS spectra for both molecules obtaining the dI/dV maps of corresponding frontier orbitals for 1. Due to the three-dimensional geometry of the molecule, it was difficult to achieve clear DOS distributions to be associated to specific molecular orbitals. Moreover, the acquisition of the dI/dV maps for 2 at the corresponding peak positions led to the transformation of the molecule. Therefore, we decided to only include the STS point spectra in the SI file.

Action: STS point spectra showing the energy position of the frontier orbitals (Figure R2) have now been added to the SI file.

Figure R2. Long-range scanning tunneling spectroscopy of species **1** and **2**. a) Differential conductance spectra on selected positions of **1**; the positions at which the spectra on the molecule were acquired is highlighted in the inset STM topography image with filled blue and green circles. Open feedback parameters: $V_b = 2.3$ V, $I_t = 100$ pA, $V_{rms} = 20$ mV. b) Differential conductance spectra on selected positions of **2**. The purple and red circles highlight the positions at where the spectra were acquired, showing features assigned to the HOMO and LUMO frontier orbitals. Open feedback parameters: (positive region) $V_b = -1.0$ V, $I_t = 100$ pA, $V_{rms} = 20$ mV; (negative region) $V_b = 1.0$ V, $I_t = 100$ pA, $V_{rms} = 20$ mV. The orange circles in the inset images of (a and b) correspond to the reference dI/dV spectra acquired on Au(111).

5. Although the authors write in the Method section that the tip is placed at the center of the molecule during the tip manipulation, no indication is marked in Figure 4. The site of manipulation on molecules **1** and **2** should be clearly indicated. In addition, have the authors attempted manipulation at different positions of the molecule **1** and **2** and does their voltage thresholds change? Regardless of whether it changes or not, manipulation experiments on different positions would benefit the understanding of the manipulation mechanism.

We thank the reviewer for their useful comment which has helped us to improve the manuscript. The switching between **1** and **2** described in Figure 4 has been realized by acquiring STM images with specific scanning parameters (see i) and ii) in its figure caption). Therefore, following such protocol the tip is not located at a precise position of the molecule. We apologize if the description in the Methods was not clear enough and we have now modified the corresponding text accordingly, also in reply to previous points, highlighting the difference between the “voltage pulses” and “scanning method”.

During “voltage pulses” (i.e., bias ramping with the tip at constant height, far from the molecule, as described in the revised version of the Methods) the tip was located at the “brighter” positions of molecules **1** and **2** (observed in the STM images), which, according with DFT calculations correspond to the segments of the molecules that are adsorbed further away from the surface. These locations provide, in our opinion, a more accessible point for achieving the transformation, while we have not systematically varied the manipulation site across the molecular structure.

However, we have acquired repeated STM images varying the bias voltage (from 0.1 V to 1.4 V and from -0.2 V to -1.4V, with $I = 20$ pA) without observing the switching between **1** and **2**, suggesting that there is no voltage threshold change within the applied parameters.

Action: We have added a clarification sentence in the figure caption, the Method section and the text related to the scanning parameters used in the transformation between **1** and **2**.

- Figure caption of Figure 2: “Middle: Chemical sketch of the structures observed on the Au(111) surface upon deposition of 2H-1 and applying voltage from the STM tip (see Method section for experimental details and voltage application procedure). Arrows indicate the direction of the observed transformations and the numbers below arrows specify the voltage between the tip and the sample at which the transformation was observed (ranges arise from multiple experiments).”
- Methods section: “Regarding the voltage applications reported in Figure 2, they are not standard pulses applied within a short time frame, but performed by placing the tip on top the molecule and ramping up the bias voltage until an event occurs.”
- Main text: “[...]protruding out of the molecule (Fig. 2 middle). Note that scanning over **1** or **2** using intermediate voltage parameters (from 0.1 V to 1.4 V and from -0.2 V to -1.4 V, with $I = 20$ pA) does not give rise to any molecular transformation. An illustrative sequence of the reversible switching process[...].”

6. In Page 8, line 194, the authors mentioned that the exchange coupling energy of molecule **1** was calculated using DFT in gas phase. However, the detailed description of the calculation is not found in the Method section and Supporting Information.

We are thankful to the reviewer for pointing out this missing information.

Action: We have added the missing information to the Methods section as follows:

[...]The orbitals were extrapolated to the vacuum region in order to correct the wrong decay of the charge density due to the localized basis set. To compute the triplet/singlet energy difference in the gas phase, we used the geometry of the adsorbed molecule and unrestricted Kohn–Sham DFT, starting from a ferromagnetic (antiferromagnetic) guess and imposing a multiplicity of 3 (1). From the same geometries, we built a tight-binding model considering[...].”

Additionally, some minor issues should be noticed:

1. Page 6, line 137. The subgraphs in Figure 2 were not labeled with a, b, c....

We thank the Reviewer for noticing this inconsistency.

Action: We now refer to the respective parts of Fig. 2 as top, middle, and bottom in accordance with the Fig. 2 caption.

2. Page 8, line 176. The values of scale bars should be provided, although they are same.

In Figure 3, the image sizes change slightly.

Action: Hence, we left the scale bars at each STM image and added the value on top of the first one, to be consistent with the layout of Figure 4.

Reviewer 2

1. It remains open whether the authors have attempted the full solution-phase synthesis of the targeted structures 1 or 2 (or 8), starting from 2H-1 or 7? What experiments were conducted (conditions, outcome, observations), and did any of the attempts lead to similar side products, such as compound 9?

We thank the Reviewer for raising this point. We had not included the solution-phase results in the original manuscript, as the primary focus was on surface chemistry. However, we agree that these findings provide valuable supporting information and may be of interest to solution chemists. We have now incorporated a description of these results into the main text and added the corresponding characterization data for compound 8 (NMR, HRMS, XRD) to the Supplementary Information. Please note that due to these additions, the numbering of compounds has been updated: compounds 8 and 9 are now referred to as 9 and 10, respectively.

Action: We have added the following paragraph in the main text:

“[...]both revealing the fully assembled helical skeleton composed of eight six-membered rings.

To probe whether [8]cethrene can be generated in solution, 2H-1 was oxidised with *p*-chloranil in toluene under oxygen-free conditions, leading to the formation of a precipitate within 30 minutes. On exposure of this suspension to air, diketo compound 8 was isolated, the structure of which was confirmed by NMR spectroscopy and XRD analysis (see Supplementary Information). These observations suggest that oxidation of 2H-1 generates reactive mono- and/or diradical species, which form under the reaction conditions insoluble σ -oligomers. Upon exposure to air, these oligomers may transform into 8 either directly or via dissociation into monomeric mono- and/or diradical species. However, because the formation of free [8]cethrene in solution could not be directly confirmed and the involvement of monoradical intermediates cannot be excluded, its solution-phase generation remains inconclusive.

To investigate the electronic properties and switching behaviour of [8]cethrene, the dihydro-precursor 2H-1 was deposited[...]

2: The citation of references 39 (and 40) is misleading when describing the [7]cethrene-Me₂ case. It should be clarified that these references refer not to dimethyl[7]cethrene but to a different spin-state switch. The difference in structure and properties should also be shown in Figure 1 to give context of the current design. I admit this is the previous work of this reviewer and typically I refrain from suggesting references to my own work during the peer review process. However, it might be the most relevant context to this study.

We agree with the Reviewer that this aspect was not sufficiently clear and that featuring this switch in Figure 1 is important, as it represents the most relevant example of a solution-phase magnetic switch. We thank the Reviewer for raising this important point.

Action: We modified the main text (as well as Fig. 1) accordingly:

“[...]can be promoted by light or heat in solution (Fig. 1, middle). Although this switch did not produce a magnetic response, as the triplet state of [7]cethrene-Me₂ is thermally inaccessible at room temperature, an analogous magnetic photoswitch operating at cryogenic temperatures was later reported³⁹ by Dumele and coworkers, validating the concept (Fig. 1, middle).

Here, we present the second homolog in the cethrene series[...]

3: Previous work on spin state switching on an Ag-surface based on an organic ligand system on a metal-porphyrin was reported. The switching trigger was the injection of an electron. The paper should be cited: Nature Nanotechnology 2020, 15, 18–21. Their read-out was also based on the Kondo resonance.

We appreciate the Reviewer’s careful reading and for noting the absence of this important reference.

Action: We have mentioned this example and included the corresponding reference:

“Up until now, the on–off switching of the Kondo resonance has been achieved by covalent bonding to the surface⁴² or to a hydrogen atom⁴³, intra-⁴⁴ and intermolecular⁴⁵ conformational changes, geometrical distortion⁴⁶, magnetic field⁴⁷, electron-injection-induced coordination⁴⁸ and variation of a tip–sample distance⁴⁹”

4: The solvent system to obtain single-crystal of 2H-1 is inconsistent. In figure 2 (top right); CH₂Cl₂ and in X-ray crystallography methods CD₂Cl₂.

We thank the Reviewer for spotting this mistake. The correct solvent is CD₂Cl₂. However, since the XRD structure in Fig. 2 has been replaced with that of compound **8**, the product obtained after exposing 2H-1 to oxidant and then air, this correction is no longer necessary.

5: I highly suggest specifying the title and mentioning the “on surface” study in the title. The field of surface science is clearly distinguished and many phenomena that can be observed on surface are practically impossible to achieve in bulk solution. The current study is one such example and it should be reflected in the title, maybe as “On-surface controlled magnetic bistability of a helical non-Kekulé hydrocarbon” or “Controlled magnetic bistability of a helical non-Kekulé hydrocarbon on Au(111) surface”.

We thank the Reviewer for this suggestion, with which we fully agree. We adopted the second proposed option.

Action: The title was modified to “Controlled magnetic bistability of a helical non-Kekulé hydrocarbon on a Au(111) surface”.

Reviewer 3

1. Although the Introduction and Conclusion sections are labelled, there is no label for Results, which is very confusing in the first reading, knowing this is not a letter.

We thank the Reviewer for bringing this to our attention.

Action: The label "**Results and discussion**" was added in the main text.

2. As the authors describe in the main text, there have been many reports on the on and off switching of Kondo resonances by various means. In this respect, to really warrant sufficient novelty to this work, they should show that the switching is really controllable and at least provide a better characterization of it (e.g. switching traces or average life time as a function of bias, switching probability as a function of bias and/or current...). Right now it is extremely vaguely described, providing three "switching conditions" (3.2 V pulse, $U=-5\text{mV}/I=20\text{pA}$ scan and $U=-1.5\text{V}/I=20\text{pA}$ scan) that are neither compared nor quantitatively analyzed by any means.

We appreciate the reviewer's insightful comment. Previously reported studies on on/off Kondo resonances induced by lateral manipulation (Mishra, S., et al., Nat. Nanotechnol. 15, 81 (2020)) or tip-controlled dehydrogenation (Zhao, C., et al., Phys. Rev. Lett. 132, 046201 (2024)) describe irreversible processes, where a specific nanographene reaches a final magnetic ground state. **The key novelty of our work lies in achieving a reversible spin switch, where the system can be toggled between two magnetic states in a controlled and repeatable manner via external stimuli.** This switching is highly reproducible, as highlighted in the manuscript:

"The switching from species 1 to 2 and vice versa was experimentally tested 40 times (20 times transforming 1 into 2 and 20 times transforming 2 into 1). The switching was successfully achieved in all cases when induced during the scan."

Regarding the characterization of switching conditions, we have conducted additional experiments to deepen our understanding of the phenomenon. The corresponding modifications in the main text and supporting information have been addressed in our response to Reviewer 1 (questions 1 and 5).

3. The assignment of 1 at the herringbone elbows to $S=1/2$ is not fully convincing. The tails around the Kondo resonance actually cause an impression as if they could be made up by the same lateral side-peaks/excitations as out-of-elbow, only a little bit broader or worse resolved. The authors should discuss this option.

We appreciate the Reviewer's observation. The interpretation of $S = 1/2$ state at the herringbone elbows for 1 is based on both the experimentally obtained Kondo resonances and reinforced by several previous works of magnetic nanographenes pinned by gold at the reactive elbow sites. Though the three spectra shown in Figure 3 were acquired with identical open-feedback parameters (see Figure caption), we acknowledge that the spectral tails of the Kondo resonance could, at first glance, resemble the lateral side-peaks observed for species 1 in out-of-elbow locations. Therefore, we have performed more spectroscopic measurements of different species 1 located on the elbows observing always the same symmetric profile, consistent with a single screening channel for an $S = 1/2$ system (Figure R3).

Figure R3. Electronic characterization of species 1 on the elbow. For four different species 1 no lateral side-peaks/excitations can be discerned. Open-feedback parameters for the dI/dV spectra: a) $V_b = 50$ mV, $I_t = 0.35$ nA; root mean squared modulation voltage $V_{rms} = 0.8$ mV. b) $V_b = -1$ V, $I_t = 0.1$ nA; root mean squared modulation voltage $V_{rms} = 10$ mV. c,d) $V_b = 50$ mV, $I_t = 0.5$ nA; root mean squared modulation voltage $V_{rms} = 0.8$ mV. (a, b and c) were acquired with a CO-functionalized tip while (c) was acquired with a metal tip.

Action: We have included more references to literature where a magnetic nanographene is pinned to the gold substrate at an elbow site as well as softened our statement regarding the assignment of 1 at the herringbone elbows and now the text reads as follows:

“When 1 is located on the elbow, we observe a spectroscopic feature centered at the Fermi level, which we attribute to a narrow Kondo resonance (FWHM ~ 3 meV), consistent with an unpaired electron spin ($S = 1/2$) interacting with the electron bath of Au(111) (Fig. 3g). This finding suggests that 1 might be pinned to the elbow site of the herringbone ridge, possibly binding to an under-coordinated gold atom^{52,53,54,55}, which could lead to a reduction in the number of unpaired electrons from two to one (see Fig. 3a for the corresponding chemical sketch).”

Response to Reviewer's Comments

Reviewer 1

Remarks to the Author:

The long-range STS of species 1 and 2 were provided in the revised SI. The results reveal a significant difference in term of HOMO-LUMO energy gap and their energy level. For instance, the HOMO position shifts nearly 1 eV lower from specie 1 to specie 2. How does this experimental finding compare with the theoretical results?

Figure S1. Long-range scanning tunneling spectroscopy of species 1 and 2. a) Differential conductance spectra on selected positions of 1; the positions at which the spectra on the molecule were acquired is highlighted in the inset STM topography image with filled blue and green circles. Open feedback parameters: $V_b = 2.3$ V, $I_t = 100$ pA, $V_{rms} = 20$ mV. b) Differential conductance spectra on selected positions of 2. The purple and red circles highlight the positions at where the spectra were acquired, showing features assigned to the HOMO and LUMO frontier orbitals. Open feedback parameters: (positive region) $V_b = -1.0$ V, $I_t = 100$ pA, $V_{rms} = 20$ mV; (negative region) $V_b = 1.0$ V, $I_t = 100$ pA, $V_{rms} = 20$ mV. The orange circles in the inset images of (a and b) correspond to the reference dI/dV spectra acquired on Au(111).

Reply:

We thank the reviewer for pointing out the ambiguity in the labeling of Figure S22. The significant difference in the observed energy gaps arises from their fundamentally different physical origins. For species **2**, the gap corresponds to the conventional HOMO–LUMO gap in closed-shell molecules, determined by the hybridization of the two zero modes (see Figure R1b). In contrast, species **1** is an open-shell system with two ferromagnetically coupled spins in two singly occupied molecular orbitals (SOMOs). In this case, the gap is not governed by hybridization but instead by the Coulomb gap, which must be overcome to add an additional electron to a SOMO (see Figure R1a). Theoretical results confirm that the Coulomb gap is smaller than the HOMO–LUMO gap. In the experiments, we have attempted measuring LDOS maps at relevant bias values, but the bulky and 3D structure of the molecules prevented a clear distinction of the specific orbitals in play. However, according to the new theoretical insights, we can assign the states observed in the experimental long-range STS spectra. For species **1**, the peak observed at about -0.5 V (previously assigned to HOMO) is now identified as SOMO. The SOMO is not visible, as sometimes experienced in similar systems (K. Biswas, *et al. Nanomaterials* **2022**, 12, 224; S. Mishra, *et al. Nat. Chem.* **2021**, 13, 581–586; A. Sanchez-Grande, *et al. Angew. Chem. Int. Ed.* **2020**, 59, 17594–17599). The state observed at $+2.1$ V (previously assigned to LUMO) is assigned to the LUMO+1. For species **2**, we confirm the correct assignment of HOMO and LUMO for the states at -1.3 V and $+1.4$ V, respectively.

Figure R1: Energy levels of molecule **1** (a) and molecule **2** (b), obtained for the ground state solution of the corresponding MFH models (see Methods for details). Solid black line denotes the Fermi level.

Action taken:

We have revised the labeling of the ion resonances in Figure S22a to "SOMO" and "LUMO+1". Moreover, we have included the new theoretical results in the same figure, for a comprehensive description of the observed spectral features. Finally, the corresponding description of the long-range STS spectra has been revised as follows:

Figure S22. Long-range scanning tunneling spectroscopy of species 1 and 2. **a.** Differential conductance spectra on selected positions of 1; the positions at which the spectra on the molecule were acquired is highlighted in the inset STM topography image with filled blue and green circles. Open feedback parameters: $V_b = 2.3$ V, $I_t = 100$ pA, $V_{rms} = 20$ mV. **b.** Differential conductance spectra on selected positions of 2. The purple and red circles highlight the positions at where the spectra were acquired. Open feedback parameters: (positive region) $V_b = -1.0$ V, $I_t = 100$ pA, $V_{rms} = 20$ mV; (negative region) $V_b = 1.0$ V, $I_t = 100$ pA, $V_{rms} = 20$ mV. The orange circles in the inset images of **a** and **b** correspond to the reference dI/dV spectra acquired on Au(111). **c,d.** Energy levels of molecule 1 (c) and molecule 2 (d), obtained for the ground state solution of the corresponding MFH models (see Methods for details). Solid black line denotes the Fermi level.

For species **2**, the gap corresponds to the conventional energy difference between the highest occupied molecular orbital (HOMO) and the lowest unoccupied molecular orbital (LUMO) (HOMO–LUMO gap) in closed-shell molecules, determined by the hybridization of the two zero modes (see panel d). In contrast, species **1** is an open-shell system with two ferromagnetically coupled spins in two singly occupied molecular orbitals (SOMOs). In this case, the gap is not governed by hybridization but instead by the Coulomb gap, which must be overcome to add an additional electron to a SOMO (see panel c). The experimental scanning tunneling spectroscopy (STS) results detect the HOMO and LUMO states for species **2** at -1.3 V and $+1.4$ V, respectively. In the case of molecule **1**, instead, we revealed the SOMO at -0.5 V and the LUMO+1 at $+2.1$ V, while the SUMO was not experimentally detected. Both the experimental and theoretical results confirm that the Coulomb gap is smaller than the HOMO–LUMO gap.